# FINE-TUNING FLOW MATCHING VIA MAXIMUM LIKE-LIHOOD ESTIMATION OF RECONSTRUCTIONS

## ABSTRACT

Flow Matching (FM) algorithm achieves remarkable results in generative tasks especially in robotic manipulation. Building upon the foundations of diffusion models, the simulation-free paradigm of FM enables simple and efficient training, but inherently introduces a train-inference gap. Specifically, we cannot assess the model's output during the training phase. In contrast, other generative models including Variational Autoencoder (VAE), Normalizing Flow and Generative Adversarial Networks (GANs) directly optimize on the reconstruction loss. Such a gap is particularly evident in scenarios that demand high precision, such as robotic manipulation. Moreover, we show that FM's over-pursuit of straight predefined paths may introduce some serious problems such as stiffness into the system. These motivate us to fine-tune FM via Maximum Likelihood Estimation of reconstructions - an approach made feasible by FM's underlying smooth ODE formulation, in contrast to the stochastic differential equations (SDEs) used in diffusion models. This paper first theoretically analyzes the relation between training loss and inference error in FM. Then we propose a method of fine-tuning FM via Maximum Likelihood Estimation of reconstructions, which includes both straightforward fine-tuning and residual-based fine-tuning approaches. Furthermore, through specifically designed architectures, the residual-based fine-tuning can incorporate the contraction property into the model, which is crucial for the model's robustness and interpretability. Experimental results in image generation and robotic manipulation verify that our method reliably improves the inference performance of FM.

## 1 INTRODUCTION

Deep generative models refer to a category of deep learning techniques designed to approximate and generate samples from an unknown underlying data distribution. A mainstream paradigm is to learn a mapping between a fixed (e.g., standard normal) distribution and the data distribution. This category notably includes diffusion models, which are the current state of the art on many gernerative modelling tasks. Particularly, they have also achieved remarkable results in robot motion generation tasks (Chi et al., 2023). The mathematical principles behind diffusion can be described by SDEs (Song et al., 2021). Naturally, we can also establish the relationship between noise and samples through ODE trajectories to simplify the model and achieve faster training and inference times. This inspired the development of the Flow Matching (FM) algorithm (Lipman et al., 2023; Liu et al., 2023; Albergo & Vanden-Eijnden, 2023). FM has garnered extensive attention, particularly emerging as the leading approach in robot policy due to its fast inference speed (Black et al., 2024; Zhang et al., 2025; Braun et al., 2024; Chisari et al., 2024; Zhang & Gienger, 2024).

FM, which inherits the characteristics of diffusion, employs a simulation-free training approach. This means that during the training phase, we only train some intermediate variables, e.g., vector filed (Lipman et al., 2023), score (Song & Ermon, 2019), and noise (or the previous state) (Ho et al., 2020). We cannot directly observe and optimize the final output from these difference or differential terms. In contrast, other generative models directly includes generated samples in their training loss. In Variational Autoencoder (VAE), we contain the rescontruction error (Kingma & Welling, 2013). In normalizing flow, we use the Maximum Likelihood Estimation (MLE) of the final output generated by the model (Rezende & Mohamed, 2015; Chen et al., 2018). Generative Adversarial Networks (GANs) are similar except they use adversarial training to replace the likelihood function (Goodfellow et al., 2014).

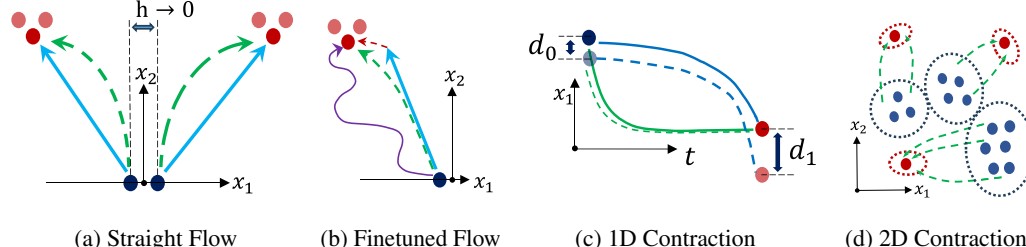

(a) Straight Flow     (b) Finetuned Flow     (c) 1D Contraction     (d) 2D Contraction

Figure 1: Fine-Tuning Your Flow: A Visual Explanation. These figures plot trajectories under different vector field models. **(a)** illustrates that over-pursuing straightness (blue lines) leads to discontinuities in the vector field, i.e., $f(0^+, 0) \neq f(0^-, 0)$. This will cause the system to exhibit stiffness, significantly exacerbating the difficulty of numerical solution and thereby compromising the model's reliability. By comparison, the green line depicts a more stable flow. **(b)** plots the pre-trained FM path (blue line), the fine-tuned flow (green line), the flow fine-tuned with residuals (red line), and the flow trained entirely with MLE (purple line). An oversimplified assumption like "the straight path" can lead to underfitting (blue line). A fine-tuned model converges to a local optimum near the pre-trained model, thereby improving its fit to the sample while maintaining path simplicity (green line). By contrast, the CNF (purple line) solely fits the samples without any prior guidance on the vector field's shape, which can easily produce overly complex trajectories. The red line utilizes a residual fine-tuning approach that preserves the pre-trained model unchanged, employing solely a residual network to learn the remaining residual components. **(c)** plots the variation curve of component $x_1$ over time. Here, compared with the blue line, the green one represents a "contracting" trajectory (Lohmiller & Slotine, 1998). When subjected to a minor disturbance $d_0$ (which may arise from stochastic noise or slight differences in external inputs), a contracting trajectory still tends to stabilize around a similar solution. Such a model demonstrates superior performance in terms of stability and robustness. **(d)** illustrates contraction in 2-dimensional space. (Blue) points in different contraction regions will converge to different (red) destinations. Points in the same contraction region behave stably and robustly.

Therefore, though offering relatively fast training speed, we have no knowledge of how the real samples are actually generated during the simulation-free training phase. This implies the existence of a gap between its training and inference phases. Such a gap can considerably impact scenarios that require high precision, such as robotic manipulation or spatio-temporal data imputation and prediction. By comparison, the Action Chunking with Transformers (ACT) architecture (Zhao et al., 2023), built upon a VAE with integrated reconstruction error, performs remarkably well in fine manipulation tasks. Moreover, we point out that FM's over-pursuit of straight paths may render the system stiff or even lead to discontinuous vector fields (see Fig. 1a), resulting in numerical instability and model unreliability. Although in practice, the stochastic noise introduced by batching and early stopping technique can smooth the model output and mitigate this phenomenon, it comes at the cost of underfitting. Fortunately, due to the smoothness of ODE trajectories (Chen et al., 2018), it is feasible to fine-tune FM directly by reconstruction error. There are multiple ways to track parameter gradients (Kidger, 2021, Chapter 5) such as adjoint sensitivity method.

The schematic diagrams in Figure 1 outline the core principles of our proposed methodology. Fine-tuning enhances the representational capacity of vector fields while preserving their simplicity. Notably, it can imbue a flow with contraction properties, leading to robust stability against minor perturbations and a latent space with higher semantic quality. This paper is organized as follows. We first theoretically analysis the relation between training loss and inferring error (Theorem 1). Then we propose a method of fine-tuning FM via MLE of reconstruction inference error (Theorem 2), which includes both straightforward fine-tuning and residual-based fine-tuning approaches. Furthermore, through specifically designed architectures, the residual-based fine-tuning can incorporate contraction properties into the model (Theorems 3-4). Experimental results verify that our method reliably improves the inference performance of FM. Our primary contributions are: (1) the first theoretical analysis quantifying the relationship between training and inference errors in FM; (2) a practical MLE-based fine-tuning framework that includes both an easy-to-implement version and a robustness-oriented residual variant with contraction analysis; and (3) comprehensive experimental validation.

## 2 RELATED WORK

**NODE and CNF**   Neural Ordinary Differential Equations (NODE), or Continuous Normalizing Flow (CNF) in the context of generation tasks (Chen et al., 2018), also adpot the ODE model. In contrast to FM's dependence on predefined reference paths, these methods utilize an optimization framework grounded in Maximum Likelihood Estimation (MLE), using the change of variables theorem with a tractable noise distribution to compute probabilities. Consequently, they tend to generate complex and intractable vector fields (see Fig. 1b). Significant efforts have been devoted to its improvement. Dupont et al. (2019) project the state into a higher-dimensional space to enable simpler paths. Finlay et al. (2020) introduce regularizations that encourage neural ODEs to prefer simpler dynamics. However, in most tasks, these methods still fail to match the simplicity and efficiency of simulation-free approaches like diffusion and FM. But their direct optimization of the end-to-end loss theoretically allows for a higher potential performance.

**Convergence and Stability**   Some work is also being done on the progressive stability analysis of networks based on ODE models. Llorente-Vidrio et al. (2021) study a class of NODEs, using the asymptotic stability to guide the design of network weights. Mei et al. (2024) investigate the theoretical conditions for convergence. These two studies are primarily focused on the task of image classification by NODE, though Mei et al. (2024) experimentally explore the ability of their network to model dynamics. The theoretical analysis of convergence, stability and contraction focus on the final steady-state characteristics of the model, rather than the preliminary dynamic behavior. Our work is for the flow model in generation tasks (one of the most prevalent approaches in this domain especially in robot manipulation). Generative models desires solution diversity, but convergence (Mei et al., 2024; 2022; Efimov & Aleksandrov, 2021) refers to a global convergent (unique) solution. Thus, in this paper we study contraction (Lohmiller & Slotine, 1998) instead, which is more concerned with the local aspect: do nearby trajectories converge to one another? This paper enables the partitioning of the domain into distinct contraction regions and then analyzes the contraction within each one. Moreover, in terms of network design, we adhere to the principle that dynamic characteristics (as long as the state remains bounded) do not affect the subsequent steady-state contraction property. We simply added a residual fine-tuning network segment at the later stage. This approach not only preserves the powerful approximation capabilities of the flow-matching component involving UNet or Transformer structures but also ensures stability and contraction in the later phases. Note that stability analyses have also been performed on NODE variants including SODEF (Kang et al., 2021), SNDEs (White et al., 2023), and Stable Neural Flows (Massaroli et al., 2020).

**Optimal-transport conditional FM**   The original FM method constructs paths based on individual sample pairs, which may lead to twisted, entangled, and complex trajectories across the overall distribution. Several studies have been dedicated to addressing this issue. Liu (2022) propose Rectified Flow, which leverages the observation that flow models follow a global predefined path yet make local adjustments to noise-sample pairings. This enables trajectory distillation but suffers from accumulated error through iteration. Tong et al. (2024) and Pooladian et al. (2023) consider the joint probability between noise and samples within a batch. By constructing optimal transport within minibatches, it approximates the optimal path between global distributions. This paper applies the principles of Tong et al. (2023; 2024) to determine the pairing between noise and samples.

**Consistency and Fast Inference**   The consistency model (Song et al., 2023) or consistency FM (Yang et al., 2024) represents another research direction aimed at achieving rapid single-step inference, albeit with a deliberate compromise on model accuracy. The Consistency Model learns to map any point on a PF-ODE trajectory directly to its origin (clean data). The reliance on distillation limits their practicality in robotics (Lu et al., 2024). Consistency flow defines a straight flow from any time step to a fixed endpoint by constraining its velocity, and has been applied in robot policy (Zhang et al., 2025). However, as previously discussed, these practices can lead to an excessively large Lipschitz constant (Fig. 1a), resulting in stiff behavior. This could pose potential risks for a system described by differential equations. Our vision is to accelerate inference speed by constructing a simple, smooth, and contracting vector field that fundamentally reduces the difficulty of numerical solutions. Furthermore, with the help of contraction analysis, we could possibly map each contraction region to its corresponding equilibrium point, thereby achieving single-step inference without relying on distillation and without loss of accuracy.

## 3 ON THE FINE-TUNING OF FM: MOTIVATIONS AND METHODS

### 3.1 THE TRAINING-INFERENCE GAP IN FM

Let $\mathbb{R}^d$ denote the data space with data points $x = (x^1, \dots, x^d) \in \mathbb{R}^d$. Denote the probability path $p_t : \mathbb{R}^d \to \mathbb{R}_+$, which is a time dependent (for $t \in [0, 1]$) probability density function, i.e., $\int p_t(x)dx = 1$, and a time-dependent vector field, $u_t : [0, 1] \times \mathbb{R}^d \to \mathbb{R}^d$. A vector field $u_t$ constructs a time-dependent diffeomorphic map, called a flow, $\psi : [0, 1] \times \mathbb{R}^d \to \mathbb{R}^d$, defined via the ordinary differential equation (ODE):

$$\frac{d}{dt}\psi_t(x(0)) = u_t(\psi_t(x(0))), \quad \psi_0(x(0)) = x_0. \tag{1}$$

Given two marginal distributions $q_0(x_0)$ and $q_1(x_1)$ for which we would like to learn a model to transport between, FM seeks to optimize the simple regression objective $\mathbb{E}_{t,p_t(x)} \|v_t(x; \theta) - u_t(x)\|^2$, where $u_t(x)$ is the is a vector field that generates a probability path $p_t$ under the two marginal constraints, $v_t(x; \theta)$ is the parametric vector field. Let $\phi_t(x_0)$ be the solution to the ODE $\frac{d}{dt}\phi_t = v_t(\phi_t; \theta)$ with initial value $\phi_0 = x_0$. To obtain the numerical solution, let us define $N$ as the total number of discrete steps, $t_i$ as the $i$-th time point, $\tau_i = t_{i+1} - t_i$ as $i$-th interval. Different time points typically involve different step sizes if we employ adaptive step size algorithms, as is often the case. We use $\hat{\phi}_n(x_0)$ to represent the corresponding numerical solution at time $t_n$, with the model error $\varepsilon_n(x_0) := \psi_{t_n}(x_0) - \hat{\phi}_n(x_0)$.

For computational tractability, we use the equivalent Conditional Flow Matching (CFM) objective $\mathcal{L}_{\text{CFM}}(\theta) = \mathbb{E}_{t,q(x_1),p_t(x|x_1)} \|v_t(x) - u_t(x|x_1)\|^2$ (Lipman et al., 2023). Reparameterizing $p_t(x|x_1)$ in terms of just $x_0$ we get

$$\mathcal{L}_{\text{CFM}} = \mathbb{E}_{t,q(x_0,x_1)} \|v_t(\psi_t(x_0|x_1); \theta) - u_t(\psi_t(x_0|x_1)|x_1)\|^2, \tag{2}$$

where $\psi_t(x_0|x_1)$ is the conditional flow with a predefined form. We typically use $\psi_t(x_0|x_1) = (1-t)x_0 + tx_1$ with the corresponding vector field $u_t(\psi_t(x_0|x_1)|x_1) = x_1 - x_0$. Thus we have

$$\mathcal{L}_{\text{CFM}} = \mathbb{E}_{t,q(x_0,x_1)} \left[\|x_0 + \Delta t v_t(t, \psi_t(x_0|x_1); \theta) - x_1\|^2\right], \quad \text{with } \Delta t = 1. \tag{3}$$

This can be seen as measuring the ground truth $x_1$ and the numerical result by implicit one-step Euler method within time interval $[0, 1]$, with derivative estimated in time $t$. We can also turn to a more advanced solving scheme, as we do during the inference phase. The difference is that when in training we use the ground truth value $\psi_t(x_0|x_1)$ since we have predefined the path, but when inferring we should use the estimated value $\hat{\phi}_n(x_0)$. And this creates the gap between the training stage and inference (or prediction). Fortunately, we can bound this gap by the following theorem.

**Theorem 1** *Assume that the truth vector field $u_t(x)$ is a Lipschitz-continuous function with the Lipschitz constant $L_u > 0$. And the discrepancy between the learned vector field and the truth satisfies $\|v_\theta(t, x) - u_t(x)\|_\infty \leq \delta$, then we can derive the following error estimate between the ground-truth values and the network's inferred values*

$$|\varepsilon_N| \leq \exp(L_u t_{N-1}) \left(\Sigma_{j=0}^{N-1}(\delta \tau_j + \frac{1}{2}M\tau_j^2) + |\varepsilon_0|\right). \tag{4}$$

*where $M = \max_{0 \leq t \leq 1} |\ddot{\psi}_t|$ is an upper bound for the second time derivative. Under special circumstances, when uniform step sizes are adopted, we obtain a more refined estimation formula,*

$$|\varepsilon_N| \leq \exp(L_u)\varepsilon_0 + \frac{M\tau_0 + 2\delta}{2L}(\exp(L_u) - 1). \tag{5}$$

Proof is in Appendix B.

**Remark 1** *The variable step-size method, while often more efficient, introduces a degree of uncontrollability due to its excessive degrees of freedom, thereby raising the upper bound.*

**Remark 2** *Theorem 1 generalizes classical numerical analysis to settings in which the underlying vector field is approximated (e.g., via learning), introducing inherent approximation errors.*

Though this shows that the gap between training and inference is bounded, such a gap will inevitably compromise the model's effectiveness. This theorem indicates that the error in the final generated sample (or action in robot policy) $\varepsilon_N$ will further amplify the training error $\delta$, at least by a multiplicative factor $\exp(L_u)$ with an additive constant. This issue becomes particularly severe when predefined paths may cause discontinuities in the vector field (Fig. 1a), resulting in a large Lipschitz constant $L_u$. This motivates our pursuit of a consistent training-inference paradigm that can further optimaize $\varepsilon_N$. More specifically, we utilize MLE for fine-tuning based on reconstruction results, and the detailed principles and procedures will be thoroughly discussed in the following subsection.

## 3.2 MLE Fine-tuning

We begin by making the following assumption regarding the conditional distribution of given samples.

**Assumption 1** *Suppose that given the sample $x_1$, the underlying conditional distribution is a Gaussian distribution $p_1(x|x_1) = \mathcal{N}(x|x_1, \Sigma)$, where $\Sigma$ is a d-dimensional covariance matrix.*

The following theorem provides a concrete loss function for MLE fine-tuning.

**Theorem 2** *Under Assumption 1, and when $\Sigma$ is a scalar matrix, performing maximum likelihood estimation (MLE) by maximizing the expectation $\mathbb{E}_{q(x_0,x_1)}\left[\log p_1\left(\hat{\phi}_N(x_0)|x_1\right)\right]$ is equivalent to minimizing the following loss function*

$$\mathcal{L}_{\text{MLE}} = \mathbb{E}_{q(x_0,x_1)}\left[\|\varepsilon_N(x_0|x_1)\|^2\right], \tag{6}$$

*where $\varepsilon_n(x_0|x_1) := \psi_{t_n}(x_0|x_1) - \hat{\phi}_n(x_0)$ is the conditional model error.*

The proof, along with the loss function in the more general case of a diagonal matrix $\Sigma$, is given in Appendix B. (6) provides us with a computationally feasible loss function that we can directly use to fine-tune our model. As established by Theorem 2, our method enables the direct optimization of the model error $\varepsilon_N(x_0)$. This stands in contrast to previous approaches ((4) and (5) in Theorem 1), which could only provide a loose upper bound that contained non-optimizable components.

MLE by (6) has the advantage of high-precision for it directly optimizing the object obtained by inference procedure. But it suffers from several critical issues. Firstly, it is particularly prone to overfitting. Since no additional constraints are imposed on the vector field or trajectory shapes, it often generates sophisticated flow fields with convoluted solution paths (Finlay et al., 2020). This compromises the reliability of numerical solutions, resulting in highly unstable model outputs. Secondly, it is computationally expensive compared to the original FM training algorithm, for it needs repeated simulation of the ODE. These issues are major obstacles to the adoption of ODE-based models trained with MLE.

**However, we contend that MLE is particularly well-suited for fine-tuning pre-trained flow models**. The reasons are listed as follows. After the training of FM, we already get a relatively straight base model, MLE method will only fine-tune it. Therefore, the vector fields will improves accuracy without significant shape distortion. From an optimization perspective, it more readily converges to local optima near a 'straight flow'. Moreover, this convergence process of parameters is significantly faster and more efficient than training a flow model from scratch. Thus the higher computational complexity of MLE is acceptable. We emphasize that the complexity during training is generally inconsequential, as our primary focus remains inference speed which is unaffected by these training-phase design choices.

There is another fundamental aspect regarding straightness of flow we want to clarify. One might question whether fine-tuning a flow model would compromise its 'straighter' trajectory property, potentially increasing numerical solution difficulty and computational overhead during both training and inference phases. However, in practice, the actual situation may differ and the following issues may arise. First, straighter lines between sample points does not necessarily mean that the path between distributions will also be straighter (Gao et al., 2024). Second, as in Fig. 1a , excessive pursuit of straightness may lead to system stiffness or even discontinuity. This would severely hinder the numerical solution of this ODE system, which is unacceptable both in training and inference. Acutally, during the training of a flow, we often inject noise or adopt early stopping—this can also be interpreted as a form of smoothing for the system's vector field. Therefore, defining a straight

trajectory does not guarantee enabling single-step inference. The implementation of usable single-step inferring often necessitates iterative distillation, a process that introduces cumulative model errors.

More discussion of the advantages of the MLE-based training approach using model reconstruction in high-precision scenarios (e.g., robotic manipulation) is provided in Appendix C.

### 3.3 RESIDUAL MLE FINE-TUNING WITH CONTRACTION ANALYSIS

In this part we introduce a residual learning framework for fine-tuning FM robot policy, motivated by Yuan et al. (2025) and Jiang et al. (2025). They found that fine-tuning the policy by learning residuals yielded highly effective results. For our scenario, we implement it through extending the time horizon for solving the flow model from $[0, 1]$ to $[0, 1 + T]$ $(T > 0)$. Specifically, the residual part takes $\phi_1(x_0)$ as input, and generate $\phi_{1+T}(x_0)$ as the final result, i.e., $\frac{d}{dt}\phi_t(x) = \tilde{v}_t(\phi_t(x); \tilde{\theta})$ with given $\phi_1(x_0)$ for $t \in [1, 1 + T]$. Here $\tilde{v}$ is the vector field of the residual part with parameters $\tilde{\theta}$. $\tilde{v}$ is also trained via MLE loss (6) for better fitting capability, and $\tilde{v}$ features a simpler structure and fewer parameters compared to $v$ to prevent overfitting.

**ISS and Contraction** Moreover, we employ networks with specific architectures to obtain certain desirable properties, such as Input-to-State Stability (ISS) and contraction (Figs. 1c-1d). ISS refers to the property that the model output remains stable and bounded in the presence of external inputs, thereby avoiding divergence behavior as illustrated by the blue curves in Fig. 1c. ISS property is particularly crucial in fields such as robotic manipulation. We obviously do not want minor disturbances to be excessively amplified, causing severe jitter in movements. The precise definition of ISS is provided in Appendix A. Furthermore, the contraction property indicates that the model exhibits robustness against small disturbance noises, as shown by the green curves in Fig. 1c and Fig. 1d. Each contraction region can correspond to a specific semantic meaning in the latent space. We may even achieve single-step inference without relying on distillation and without loss of accuracy by directly mapping each contraction region to its corresponding equilibrium point.

Specifically, we consider the ControlSynth Neural ODE (Mei et al., 2024) or Persidskii system (Efimov & Aleksandrov, 2021; Mei et al., 2022),

$$\dot{x}(t) = A_0 x(t) + \sum_{j=1}^{M} A_j f_j(W_j x(t)) + g(u(t)), \tag{7}$$

where $x_t := x(t) \in \mathbb{R}^n$ is the robot's state vector typically including its 3-dimensional position and orientation; the matrices $A.$ are with approximate dimensions; $W.$ are weight matrices; the visual image or state input $u_t := u(t) \in U \subset \mathbb{R}^m$, $u \in \mathscr{L}_\infty^m$; $f_j = [f_j^1 \ldots f_j^{k_j}]^\top$ $(f_j : \mathbb{R}^{k_j} \to \mathbb{R}^{k_j})$ and $g : U \to \mathbb{R}^n$ ensuring the existence of the solutions of the neural network (NN) (7) at least locally in time, and $g = [g_1 \ldots g_n]^\top$; w.l.o.g., the time $t$ is set as $t \geq 0$. The definitions of the relevant symbols and further details are provided in Appendix A. Suppose that the nonlieanr function $f$ satisfying the following conditions.

**Assumption 2** For any $i \in \{1, \ldots, k_j\}$ and $j \in \{1, \ldots, M\}$, $sf_j^s(s) > 0$ for all $s \in \mathbb{R} \backslash \{0\}$.

**Assumption 3** Assume that the functions $f_j^i$ are continuous and strictly increasing for any $i \in \{1, \ldots, k_j\}$ and $j \in \{1, \ldots, M\}$.

Assumption 2 applies to many activation functions, such as $\tanh$ and parametric ReLU. With a reordering of nonlinearities and their decomposition, there exists an index $\omega \in \{0, \ldots, M\}$ such that for all $1 \leq s \leq \omega$ and $1 \leq i \leq k_s$, $\lim_{\nu \to \pm\infty} f_s^i(\nu) = \pm\infty$. Also, there exists $\zeta \in \{\omega, \ldots, M\}$ such that for all $1 \leq s \leq \zeta$, $1 \leq i \leq k_s$, we have $\lim_{\nu \to \pm\infty} \int_0^\nu f_s^i(r) dr = +\infty$. First we introduce ISS theorem.

**Theorem 3** Let Assumptions 2-3 be satisfied. If there exist positive semidefinite symmetric matrices $P$; positive semidefinite diagonal matrices $\{\Lambda^i = diag(\Lambda_1^i, \ldots, \Lambda_n^i)\}_{i=1}^M$, $\{\Xi^s\}_{s=0}^M$, $\{\Upsilon_{s,r}\}_{0 \leq s < r \leq M}$; positive definite symmetric matrix $\Phi$ such that the following linear matrix inequalities hold true:

$$P + \sum_{j=1}^{\zeta} \Lambda^j > 0; \quad Q = Q^T \leq 0; \quad \sum_{j=1}^{M} \Upsilon_{0,j} + \sum_{s=1}^{\omega} \Xi^s + \sum_{s=1}^{\omega} \sum_{r=s+1}^{\omega} \Upsilon_{s,r} > 0. \tag{8}$$

*where*

$$Q_{1,1} = A_0^\top P + PA_0 + \Xi^0; \quad Q_{j+1,j+1} = A_j^\top W_j^\top \Lambda^j + \Lambda^j W_j A_j + \Xi^j;$$

$$Q_{1,j+1} = PA_j + A_0^\top W_j^\top \Lambda^j + W_j^\top \Upsilon_{0,j}; \ Q_{s+1,r+1} = A_s^\top W_r^\top \Lambda^r + \Lambda^s W_s A_r + W_s^\top W_r \Upsilon_{s,r} W_r^\top W_s;$$

$$Q_{1,M+2} = P; \quad Q_{M+2,M+2} = -\Phi; \quad Q_{j+1,M+2} = \Lambda^j W_j.$$

*then system (7) is ISS.*

Proof is in Appendix B. Next, to analyse contraction, we consider another trajectory of the model (7) $\dot{y}(t) = \sum_{j=1}^M A_j f_j(W_j y(t)) + g(u(t))$ with the same input but different initial conditions $y(0) \in \mathbb{R}^n$. Let $\xi := y - x$. Then the corresponding error system is

$$\dot{\xi} = A_0 p_j(\xi) + \sum_{j=1}^M A_j p_j(x, \xi), \tag{9}$$

where $p_j(x, \xi) = f_j(W_j(\xi + x)) - f_j(W_j x)$. Note that for any fixed $x \in \mathbb{R}^n$, the functions $p_j$ in the variable $\xi \in \mathbb{R}^n$ satisfy the properties in Assumptions 1, 2 (with a different Lipschitz constant).

**Theorem 4** *Let Assumptions 2-3 and conditions in Theorem 3 be satisfied, and in the bounded domain (determined by the ISS property), the functions $f_j^i$ are Lipschitz continuous with Lipschitz constants $L_j^i$. If there exist positive semidefinite symmetric matrices $\tilde{P}$; positive semidefinite diagonal matrices $\{\tilde{\Lambda}^i = diag(\tilde{\Lambda}_1^i, \ldots, \tilde{\Lambda}_n^i)\}_{i=1}^M$, $\{\tilde{\Upsilon}_{j,r}\}_{j,r=1}^M$, $\{\Gamma_j\}_{j=1}^M$, $\{\Omega_j\}_{j=1}^M$; positive definite symmetric matrix $\Phi$; and positive scalars $\gamma, \theta$ such that the following linear matrix inequalities hold true:*

$$\tilde{Q} = \tilde{Q}^T \leq 0; \quad \Gamma_j - \gamma L^j \geq 0; \quad \Omega_j - \theta L^j \geq 0;$$

$$\sum_{j=1}^M \left( \Gamma_j - \gamma L^j + \Omega_j - \theta L^j \right) + \sum_{j=1}^M \sum_{r=1}^M \tilde{\Upsilon}_{j,r} > 0, \tag{10}$$

*where*

$$\tilde{Q}_{1,1} = A_0^\top \tilde{P} + PA_0 + \tilde{\Xi}^0; \quad \tilde{Q}_{2,2} = -2\gamma I; \quad \tilde{Q}_{1,2} = PA + \Gamma; \quad \tilde{Q}_{1,3} = A_0^\top \Delta + \Omega;$$

$$\tilde{Q}_{2,3} = A^\top \Delta + \tilde{\Upsilon}; \quad \tilde{Q}_{3,3} = -2\theta I; \quad A = \begin{bmatrix} A_1 & \cdots & A_M \end{bmatrix}; \quad \Gamma = \begin{bmatrix} W_1^\top \Gamma_1 & \cdots & W_M^\top \Gamma_M \end{bmatrix};$$

$$\Delta = \begin{bmatrix} W_1^\top \Lambda^1 & \cdots & W_M^\top \Lambda^M \end{bmatrix}; \ \Omega = \begin{bmatrix} W_1^\top \Omega_1 & \cdots & W_M^\top \Omega_M \end{bmatrix}; \ \tilde{\Upsilon} = (W_j^\top W_j \tilde{\Upsilon}_{j,r} W_r^\top W_r)_{j,r=1}^M,$$

*then system (7) (with trajectory $x$) is contracting. If we define $\tilde{V}(\zeta) = \zeta^\top \tilde{P}\zeta + 2\sum_{j=1}^M \sum_{i=1}^{k_j} \tilde{\Lambda}_i^j \int_0^{W_j^i \zeta} f_j^i(s)ds$, then the contraction region of $x_0$ contains $\{x_0 + \xi | V(\xi) \leq \max_{\zeta \in \mathbb{R}^d} \tilde{V}(\zeta)\}$*

Proof is in Appendix B. This theorem provides the conditions for contraction and delineates the contraction region. Theorems 3 and 4 establish the stability and contraction property for systems of the form (7) via tractable linear inequalities. In practical applications, we can strictly embed these conditions into the training process, for instance, by incorporating physics-informed loss functions. Alternatively, we may simply guide the model to select matrices $A$. with predominantly negative eigenvalues, while using these conditions for theoretical guarantees and interpretability analysis. Certainly, we can also employ networks with more complex architectures (e.g., UNet and Transformer). For the contraction analysis of these more general forms, we can refer to Lohmiller & Slotine (1998) and Li et al. (2025), albeit at the expense of increased condition complexity.

## 4 EXPERIMENTS

In this section we experimentally evaluate the benefits of fine-tuning FM. All experiment details can be found in Appendix D.

### 4.1 PRELIMINARY EXPERIMENTS

We first perform an experiment on unconditional CIFAR-10 generation from a Gaussian source to test the basic principles of our method. The baseline models are chosen as DDPM (Ho et al., 2020),

Table 1: FID score and number of function evaluations (NFE) for different ODE solvers: fixed-step Euler integration with 100 and 1000 steps and adaptive integration (Hairer et al., 1993, DOPRI5). The adaptive solver is significantly better than the Euler solver in fewer steps. First results are from Lipman et al. (2023) and the next three from Tong et al. (2024). The two last rows report the results of our fine-tuned FM. Here ResFT-FM denotes a residual MLE fine-tuning in Section 3.3 using a simplified UNet.

| NFE / sample → | 100 | 1000 | Adaptive | |
|---|---|---|---|---|
| Algorithm ↓ | FID | FID | FID | NFE |
| DDPM | | | 7.48 | 274 |
| VP-FM | 7.772 | 4.048 | 4.335 | 525.92 |
| OT-FM | 4.640 | 3.822 | 3.655 | 143.00 |
| S.I. | 4.488 | 4.132 | 4.009 | 146.12 |
| I-CFM | 4.461 | 3.643 | 3.659 | 146.42 |
| OT-CFM | **4.443** | 3.741 | 3.577 | **133.94** |
| FT-FM (ours) | 4.451 | **3.620** | **3.496** | 146.42 |
| ResFT-FM (ours) | | | 3.553 | 460 .07 |

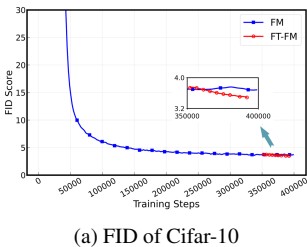

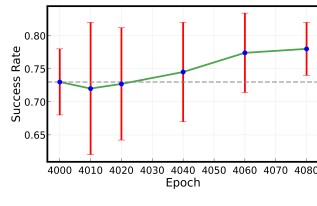

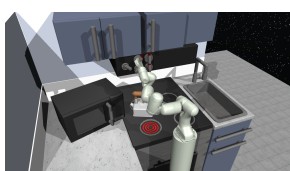

(a) FID of Cifar-10      (b) SR in Franka Kitchen      (c) Turn on Left Burner

Figure 2: Visualization of Experimental Results. **(a)** plots the FID of FM (blue curve) under different training steps. The fine-tuned The red line is the fine-tuned FID score from checkpoint at $3.5 \times 10^5$ step under the same time consumption. **(b)** plots the success rate (SR) of the fine-tuned policy from checkpoint at $4000$ epoch. The red vertical bars represent the variance of SR, which gradually decreases during the training process. **(c)** shows the task of turning on the left burner in the Franka Kitchen environment, executed by our fine-tuned FM policy within the MuJoCo simulator.

FM with Variance Exploding (VE) path and Optimal Transport (OT) path (Lipman et al., 2023), and Stochastic Interpolants (SI) (Albergo et al., 2023). We train our fine-tuned FM and report the Fréchet inception distance (FID) in Table 1. The FID over training time is in Fig 2a.

From Table 1, we can see that fine-tuning FM can improve the performance on FID with almost the same number of function evaluations (NFE). In Fig. 2a, the FID of FM model has stopped decreasing by the 350,000th step. But when fine-tune the checkpoint at that step, the FID score was further reduced since MLE enables more powerful representations. More visualization results and analysis are in the appendix. These demonstrate the significance of fine-tuning FM.

## 4.2 ROBOTIC MANIPULATION

In this part we investigate the performance of our method on three robot manipulation datasets which includes closed-loop 6D robot actions and gripper actions: Franka Kitchen (Gupta et al., 2020), push-T (Florence et al., 2022), and Robomimic (Mandlekar et al., 2022).

- Push-T involves manipulating a T-shaped block to a designated target using a circular end-effector. The policy takes RGB images along with end-effector proprioception as input and produces closed-loop end-effector actions. The task is supported by a dataset of 200 human demonstrations.

- The Franka Kitchen environment contains 7 interactive objects with 566 human demonstration sequences. Each demonstration completes any 4 tasks in variable order, and the goal is to accomplish as many tasks as possible regardless of sequence. The policy uses state-based observations and generates closed-loop commands for both robot joint movements and gripper actions.

- Robomimic offers 5 different tasks with high-quality human teleoperation demonstrations. This study specifically utilizes the Transport task, containing 200 demonstrations. The policy operates on state-based inputs and outputs closed-loop control signals for robot joints and the gripper.

| Methods (16-step) | Push-T[a] ↑ | Push-T[b] ↑ | Franka Kitchen ↑ | Robominic ↑ |
|---|---|---|---|---|
| DDPM | 0.8840/0.7178 | 0.7360/0.6100 | 0.9840/0.6716 | 0.9359/0.7168 |
| DDIM | 0.8801/0.6372 | 0.7490/0.6167 | 0.9865/0.7471 | 0.9334/0.7073 |
| FM | 0.9035/0.7519 | 0.7363/0.6218 | 0.9960/0.7425 | 0.9360/0.7289 |
| FT-FM (ours) | **0.9197/0.7885** | **0.7567**/0.6496 | **0.9967**/0.7822 | **0.9401/0.7552** |
| ResFT-FM FM[c] (ours) | 0.9143/0.7761 | 0.7452/**0.6511** | 0.9963/**0.7836** | 0.9385/0.7447 |
| ResFT-FM FM[d] (ablation) | 0.9039/0.7518 | 0.7371/0.6220 | 0.9961/0.7541 | 0.9362/0.7283 |

[a] sampling range: $[(50, 450), (50, 450), (200, 300), (200, 300), (-\pi, \pi)]$
[b] sampling range: $[(50, 450), (50, 450), (100, 400), (100, 400), (-\pi, \pi)]$
[c] fine-tuning using ControlSynth Neural ODE (7)
[d] fine-tuning using the standard conditional UNet

Table 2: We report robot performance as (max) / (average over last checkpoint with 10 replications), each averaged across 500 environment initializations. Success rate is used for all tasks except Push-T, which uses target area coverage. For Push-T, we vary initial end-effector and T-block poses. Results are shown for the Transport task in the Robomimic benchmark.

We learn the robot policy from expert data and evaluate it in the corresponding simulation environment in MuJoCo or Gym. We choose DDPM, DDIM and orginal FM as baseline models. The evaluation in each environment has been carried out across 500 different initial conditions. The baseline models are trained for 4500 epochs. The fine-tuned training starts from the checkpoint at 3500 epochs and runs for approximately 100 epochs to ensure a roughly consistent training duration. To achieve the desired contraction property, we incorporate a physics-informed loss term $\lambda_\omega \mathrm{ReLU}(\omega)$ where $\omega = \sum_{i,k} \left( |A_i^{k,k}| - \sum_{l!=k} |A_i^{k,l}| \right) / \left( \sum_l |A_i^{k,l}| + \epsilon_A \right)$ in (7) that encourages the matrix to be negatively diagonally dominant, thereby promoting a greater number of negative eigenvalues then satisfying (8) and (10). For ablation study, instead of a structurally simplistic network (7), we utilize the standard conditional UNet from the pre-training stage and apply MLE to learn the residual components. The results are recorded in Table 2 and Fig. 2b.

Table 2 reflects that fine-tuning can effectively enhance FM's performance. Ablation results show that training a complex network directly with MLE does not yield satisfactory results due to the optimization challenges posed by an excessive number of free parameters. Conversely, utilizing a structurally simpler contracting network (7) results in substantially improved stability and training efficiency. In subsequent work, we will analyze the various contraction regions and attempt to use them to accelerate inference. Figure 2b shows the success rate (SR) after fine-tuning the FM checkpoint at 4000 epochs, where the model's performance had reached its ceiling. We observe a significant improvement in the model's SR after only 80 epochs. The observed variance reduction also indicates enhanced reliability of our fine-tuning strategy.

## 5 CONCLUSION AND FUTURE WORK

While the simulation-free optimization is simple and efficient, this paper identifies some of its inherent limitations and proposes a methodology that leverages fine-tuning to further optimize FM. Experimental results robustly demonstrate the efficacy of our fine-tuning strategy. This paper also provides several novel analytical tools for flow models. We generalizes classical numerical analysis to flow settings in which the underlying vector field is approximated via learning. We combine network designs inspired from control theory to incorporate contraction property into the FM model.

In future work, we will expand the evaluation of fine-tuned FM models across a broader range of experimental settings, including spatio-temporal data forecasting. Additionally, we plan to develop methods for efficiently incorporating LMI boundedness conditions during training. The effects of more advanced numerical schemes on flow-based inference under more quantitative forms for characterizing vector field errors besides the infinity norm will be examined both theoretically and empirically. We will also explore more fine-tuning techniques, such as LoRA, and investigate alternative structured learning systems to endow the model with desirable properties such as contraction and stability. We further propose to extend this training paradigm—combining velocity matching pre-training with reconstruction error fine-tuning—to the modeling of ODE dynamical systems.

## ETHICS AND REPRODUCIBILITY STATEMENT

**Ethics statement** This work presents a theoretical and methodological advancement in generative models, especially in robotic manipulation applications. It is evaluated on standard public datasets (CIFAR-10, Push-T, Franka Kitchen, Robomimic) and does not raise any immediate ethical concerns. We encourage responsible use of the technology.

**Reproducibility Statement** We provide the following to ensure reproducibility:

- **Code**: Our code is in the accompanying supplementary material.
- **Experiments**: Our implementation is based on PyTorch 1.12.1 and Python 3.9. We include full training and evaluation scripts, and all hyperparameters are detailed in Section D.
- **Datasets**: We use the public datasets (CIFAR-10, Push-T, Franka Kitchen, Robomimic).
- **Resources**: The experiments were conducted on $8 \times$ NVIDIA A100 GPUs.

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

# APPENDIX

## A    NOTATION AND DEFINITIONS

The symbol $\mathbb{R}$ represents the set of real numbers, $\mathbb{R}_+ = \{\ell \in \mathbb{R} : \ell \geq 0\}$, and $\mathbb{R}^n$ denotes the vector space of $n$-tuple of real numbers. The transpose of a matrix $A \in \mathbb{R}^{n \times n}$ is denoted by $A^\top$. Let $I$ stand for the identity matrix. The symbol $\|\cdot\|$ refers to the Euclidean norm on $\mathbb{R}^n$.

For a Lebesgue measurable function $u \colon \mathbb{R} \to \mathbb{R}^q$, define the norm $\|u\|_{(t_1, t_2)} = \operatorname{ess\,sup}_{t \in (t_1, t_2)} \|u(t)\|$ for $(t_1, t_2) \subseteq \mathbb{R}$. We denote by $\mathscr{L}_\infty^q$ the space of functions $u$ with $\|u\|_\infty := \|u\|_{(-\infty, +\infty)} < +\infty$.

A continuous function $\alpha : \mathbb{R}_+ \to \mathbb{R}_+$ belongs to class $\mathscr{K}$ if it is strictly increasing and $\alpha(0) = 0$, and $\mathscr{K}_\infty$ means that $\alpha$ is also unbounded. A continuous function $\beta : \mathbb{R}_+ \times \mathbb{R}_+ \to \mathbb{R}_+$ belongs to class $\mathscr{K}\mathscr{L}$ if $\beta(\cdot, r) \in \mathscr{K}$ and $\beta(r, \cdot)$ is a decreasing to zero function for any fixed $r > 0$.

For this system (7), we have the following assumption.

**Assumption A.1** *We assume that the system (7) is forward complete, i.e., for all $x_0 \in \mathbb{R}^n$ and $u \in \mathcal{L}_\infty^q$, the solution $x(t, x_0, u)$ is uniquely defined for all $t \in \mathbb{R}_+$.*

The formal definitions of ISS and convergence are provided below.

**Definition A.1** *A forward complete system (7) is input-to-state stable (ISS) if there exist $\beta \in \mathscr{K}\mathscr{L}$ and $\gamma \in \mathscr{K}\mathscr{L}$ such that*

$$\|x(t, x_0, u)\| \leq \beta(\|x_0\|, t) + \gamma(\|u\|_\infty), \quad \forall t \in \mathbb{R}_+,$$

*for any $x_0 \in \mathbb{R}^n$ and $u \in \mathcal{L}_\infty^q$. The model (7) is convergent if it admits a unique bounded solution for $t \in \mathbb{R}$ that is globally asymptotically stable (GAS).*

## B    THEOREM PROOFS

### B.1    PROOF OF THEOREM 1

*To be clear and concise, we define and rearrange some notations. For an initial value $x_0$, let $x(t_n) := \psi_t(x_0)$ and $x_n := \phi_n(x_0)$ represent the ground truth and the model output, respectively. Then we can derive that*

$$\varepsilon_{n+1} = x(t_{n+1}) - x_{n+1}$$

$$= x(t_n) + \int_{t_n}^{t_{n+1}} u_\tau(x(\tau)) \, d\tau - x_n - v_{t_n}(x_n)$$

$$= \varepsilon_{n-1} + \int_{t_n}^{t_{n+1}} u_\tau(x(\tau)) - v_{t_n}(x_n) \, d\tau$$

$$= \varepsilon_{n-1} + \int_{t_n}^{t_{n+1}} f(\tau, x(\tau)) - u_{t_n}(x(t_n)) + u_{t_n}(x(t_n)) - u_{t_n}(x_n) + u_{t_n}(x_n) - v_{t_n}(x_n) \, d\tau.$$

$$\text{(B.1)}$$

We will split the integral in expression (B.1) into three parts and estimate them separately. The first part

$$
\left| \int_{t_n}^{t_{n+1}} u_\tau(x(\tau)) - u_{t_n}(x(t_n)) \, d\tau \right|
$$

$$
= \left| \int_{t_n}^{t_{n+1}} x'(\tau) - x'(t_n) \, d\tau \right|
$$

$$
= \left| \int_{t_n}^{t_{n+1}} x''(t_n + \theta(\tau - t_n))(\tau - t_n) \, d\tau \right| \qquad \text{(B.2)}
$$

$$
= \left| x''(t_n + \theta(\tau - t_n)) \int_{t_n}^{t_{n+1}} (\tau - t_n) \, d\tau \right|
$$

$$
= \left| \frac{1}{2} \tau_n^2 x''(t_n + \theta(\bar{t} - t_n)) \right|
$$

$$
\leq \frac{1}{2} M \tau_n^2,
$$

where $0 < \theta < 1$, $\bar{t} \in (t_n, t_{n+1})$, $M = \max_{0 \leq t \leq 1} |x''(t)|$. The second and third equalities in (B.2) use the Differentiation Mean Value Theorem (MVT) and Integration MVT, respectively. Using the Lipschitz condition, the second part in (B.1) derives

$$
\left| \int_{t_n}^{t_{n+1}} u_{t_n}(x(t_n)) - u_{t_n}(x_n) \right| = \tau_n \left| (u_{t_n}(x(t_n)) - u_{t_n}(x_n)) \right| \leq L_u \tau_n |x(t_n) - x_n| = L_u \tau_n \varepsilon_n.
$$

The third part $\left| \int_{t_n}^{t_{n+1}} u_{t_n}(x_n) - v_{t_n}(x_n) d\tau \right| \leq \delta \tau_n$. Plugging these into (B.1) and taking the absolute values yield

$$
|\varepsilon_{n+1}| \leq (1 + L_u \tau_n)|\varepsilon_n| + \delta \tau_n + \frac{1}{2} M \tau_n^2, \qquad \text{(B.3)}
$$

or equivalently,

$$
|\varepsilon_{n+1}| - |\varepsilon_n| \leq L_u \tau_n |\varepsilon_n| + \delta \tau_n + \frac{1}{2} M \tau_n^2. \qquad \text{(B.4)}
$$

Summing $n$ in (B.4) from $n = 0$ to $m - 1$, we have

$$
|\varepsilon_m| \leq L_u \Sigma_{j=0}^{m-1} \tau_j |\varepsilon_j| + \Sigma_{j=0}^{m-1} (\delta \tau_j + \frac{1}{2} M \tau_j^2) + |\varepsilon_0|.
$$

Appling Grönwall inequality (Lemma 1) we get

$$
|\varepsilon_m| \leq \exp(L_u t_{m-1}) \left( \Sigma_{j=0}^{m-1}(\delta \tau_j + \frac{1}{2} M \tau_j^2) + |\varepsilon_0| \right).
$$

By taking $m = N$, we obtain the final result in (4). It is noted that Lemma 1 employs specialized scaling techniques designed for variable step-size schemes. On a uniform grid, i.e., $\forall\, i \in [0, N]$, $\tau_i = \tau_0$, we can have a shaper error estimate. It follows directly from (B.3) that

$$
\begin{aligned}
|\varepsilon_{n+1}| &\leq (1 + \tau_0 L_u)|\varepsilon_n| + R \\
&= (1 + \tau_0 L_u)^2 |\varepsilon_{n-1}| + (1 + \tau_0 L_u)R + R \\
&\leq \cdots \\
&\leq (1 + \tau_0 L_u)^{n+1} |\varepsilon_0| + \left[ (1 + \tau_0 L_u)^n + (1 + \tau_0 L_u)^{n-1} + \cdots + 1 \right] R,
\end{aligned}
$$

where $R := \delta \tau_0 + \frac{1}{2} M \tau_0^2$. Therefore,

$$
|\varepsilon_n| \leq (1 + \tau_0 L_u)^n |\varepsilon_0| + \left[ \sum_{j=0}^{n-1} (1 + \tau_0 L_u)^j \right] R
$$

$$
\leq (1 + \tau_0 L_u)^n |\varepsilon_0| + \frac{R}{\tau_0 L_u} \left[ (1 + \tau_0 L_u)^n - 1 \right].
$$

Considering $\exp(n\tau_0 L_u) > (1 + \tau_0 L_u)^n$, we can obtain

$$
\varepsilon_n \leq \exp(L_u \tau_0 n)\varepsilon_0 + \frac{M\tau_0 + 2\delta}{2L}(\exp(L_u \tau_0 n) - 1). \qquad \text{(B.5)}
$$

Taking $n = N$ yields the final estimate (5).

For the sake of completeness, in the following we put the standard discrete Grönwall inequality, e.g., Liao & Zhang (2021, Lemma 3.1), and its proof.

**Lemma 1** *Let $\lambda \geq 0$, the time sequences $\{\xi_k\}_{k=0}^N$ and $\{V_k\}_{k=1}^N$ be nonnegative. If*

$$V_n \leq \lambda \sum_{j=1}^{n-1} \tau_j V_j + \sum_{j=0}^{n} \xi_j \quad for\ 1 \leq n \leq N,$$

*then it holds that*

$$V_n \leq \exp(\lambda t_{n-1}) \sum_{j=0}^{n} \xi_j \quad for\ 1 \leq n \leq N.$$

**Proof.** *Under the induction hypothesis $V_j \leq \exp(\lambda t_{j-1}) \sum_{k=0}^{j} \xi_k$ for $1 \leq j \leq n-1$, the desired inequality for the index $n$ follows directly from*

$$\lambda \sum_{j=1}^{n-1} \tau_j \exp(\lambda t_{j-1}) \leq \lambda \int_0^{t_{n-1}} \exp(\lambda t)\, dt = \exp(\lambda t_{n-1}) - 1.$$

*The principle of induction completes the proof.*

### B.2 PROOF OF THEOREM 2

*Under the Assumption 1 , we have*

$$\log p_1(x|x_0, x_1) = \log \frac{\exp\left(-\frac{1}{2}(x - x_1)^T \Sigma^{-1}(x - x_1)\right)}{\sqrt{(2\pi)^d |\Sigma|}} \tag{B.6}$$

*When $\sigma = \mathrm{diag}(\sigma^1, \cdots, \sigma^d)$ is a diagonal matrix with all positive entries, it follows from B.6 that*

$$\log p_1(x|x_0, x_1) = -\frac{1}{2} \sum_{i=1}^{d} \frac{(x^i - x_1^i)^2}{\sigma^i} - \log \sqrt{(2\pi)^d \prod_{i=1}^{d} \sigma^d} \tag{B.7}$$

*Note that the second term is independent of $x$. Moreover, considering the relation $\psi_1(x_0|x_1) = x_1$ and $\varepsilon_N(x_0|x_1) = \psi_1(x_0|x_1) - \hat{\phi}_N(x_0)$, maximizing $\mathbb{E}_{q(x_0, x_1)}\left[\log p_1\left(\hat{\phi}_N(x_0)|x_1\right)\right]$ is equivalent to minimizing $\mathbb{E}_{q(x_0, x_1)}\left[\frac{1}{2}\sum_{i=1}^{d}(\varepsilon_N^i(x_0|x_1))^2/\sigma^i\right]$.*

*In the case where $\Sigma$ is a scalar matrix, i.e., $\Sigma = \sigma I$ with $\sigma \in \mathbb{R}_+$, following the same procedure and discarding the irrelevant coefficient leads to the loss function (6).*

### B.3 PROOF OF THEOREM 3

Similar proofs can be found in Mei et al. (2024; 2022); Efimov & Aleksandrov (2021). Here, we briefly outline the general process.

**Proof of Theorem 3** *Consider a Lyapunov function*

$$V(x) = x^\top P x + 2 \sum_{j=1}^{M} \sum_{i=1}^{k_j} \Lambda_i^j \int_0^{W_j^i x} f_j^i(s) ds,$$

*where the vector $W_j^i$ is the $i$-th row of the matrix $W_j$. It is positive definite and radially unbounded due to Finsler's Lemma under the condition equation 8 and Assumption 2. Then, taking the derivative*

*of $V(x)$, one has*

$$
\dot{V} = \begin{bmatrix} x \\ f_1(W_1 x) \\ \vdots \\ f_M(W_M x) \\ g(u) \end{bmatrix}^\top Q \begin{bmatrix} x \\ f_1(W_1 x) \\ \vdots \\ f_M(W_M x) \\ g(u) \end{bmatrix} - x^\top \Xi^0 x
$$

$$
- \sum_{j=1}^{M} f_j(W_j x)^\top \Xi^j f_j(W_j x) - 2 \sum_{j=1}^{M} x^\top W_j^\top \Upsilon_{0,j} f_j(W_j x)
$$

$$
- 2 \sum_{s=1}^{M-1} \sum_{r=s+1}^{M} f_s(W_s x)^\top W_s^\top W_s \Upsilon_{s,r} W_r^\top W_r f_r(W_r x) + g(u)^\top \Phi g(u)
$$

$$
\leq -x^\top \Xi^0 x - \sum_{j=1}^{M} f_j(W_j x)^\top \Xi^j f_j(W_j x) - 2 \sum_{j=1}^{M} x^\top W_j^\top \Upsilon_{0,j} f_j(W_j x)
$$

$$
- 2 \sum_{s=1}^{M-1} \sum_{r=s+1}^{M} f_s(W_s x)^\top W_s^\top W_s \Upsilon_{s,r} W_r^\top W_r f_r(W_r x) + g(u)^\top \Phi g(u)
$$

$$
\leq -\alpha(V) + g(u)^\top \Phi g(u),
$$

*for a function $\alpha \in \mathscr{K}_\infty$. Under (Sontag & Wang, 1995, Theorem 1), we can verify the first condition of the ISS property due to the form of $V$, and the second relation can be recovered via $V \geq \alpha^{-1} \left( 2g(u)^\top \Phi g(u) \right) \Rightarrow \dot{V} \leq -\frac{1}{2}\alpha(V)$. This means that the ISS property of the NN equation 7 is guaranteed, and so is the boundedness of its solution.*

### B.4 PROOF OF THEOREM 4

To analyze the contraction property of (9), we need the following lemma.

**Lemma 2** *Under Assumption 3, we have $p_j(x,\xi)^\top p_j(x,\xi) \leq \xi^\top W_j^\top L^j p_j(x,\xi)$ and $f_j(W_j \xi)^\top f_j(W_j \xi) \leq \xi^\top W_j^\top L^j f_j(W_j \xi)$.*

Proof. *It follows from the Lipschitz continuity Assumption 3 that $|p_j^i(x,\xi)| = |f_j^i((W_j x)^i + (W_j \xi)^i) - f_j^i((W_j x)^i)| \leq L_j^i |(W_j \xi)^i|$. Here, the superscript $i$ denotes the $i$-th component of the vector. When $(W_j x)^i \geq 0$, we have $p_j^i(x,\xi) = f_j^i((W_j x)^i + (W_j \xi)^i) - f_j^i((W_j x)^i) \geq 0$ due to the Monotonicity in Assumption 3. Then $p_j^i(x,\xi) \leq L_j^i (W_j \xi)^i$. Multiplying both sides by a non-negative number $p_j^i(x,\xi)$, we get $p_j^i(x,\xi)^2 \leq L_j^i p_j^i(x,\xi)(W_j x)^i$. When $(W_j x)^i \leq 0$, we have $p_j^i(x,\xi) = f_j^i((W_j x)^i + (W_j \xi)^i) - f_j^i((W_j x)^i) \leq 0$ due to the same Monotonicity Assumption, leading to $-p_j^i(x,\xi) \leq -L_j^i (W_j \xi)^i$. Multiplying both sides by a non-negative number $-p_j^i(x,\xi)$, we get the same result $p_j^i(x,\xi)^2 \leq L_j^i p_j^i(x,\xi)(W_j x)^i$.*

*Summing over $i$ on both sides gives $\sum_i p_j^i(x,\xi)^2 \leq \sum_i L_j^i p_j^i(x,\xi)(W_j x)^i$. Or equivalently, in a compact from, $p_j(x,\xi)^\top p_j(x,\xi) \leq (W_j x)^\top L_j p_j(x,\xi) = x^\top W_j^\top L_j p_j(x,\xi)$. This completes the proof of the first part of the lemma. By noting the relation of $f(W_j x) = f(W_j x - 0) = p_j(0,\xi)$, the second part of the lemma holds naturally.*

**Proof of Theorem 4** *Consider an positive definite function*

$$
\tilde{V}(\xi) = \xi^\top \tilde{P} \xi + 2 \sum_{j=1}^{M} \sum_{i=1}^{k_j} \tilde{\Lambda}_i^j \int_0^{W_j^i \xi} f_j^i(s) ds.
$$

*Taking the time derivative of $\tilde{V}$:*

$$
\dot{\tilde{V}} = 
\begin{bmatrix}
\xi \\
p_1(x,\xi) \\
\vdots \\
p_M(x,\xi) \\
f_1(W_1\xi) \\
\vdots \\
f_M(W_M\xi)
\end{bmatrix}^\top
\tilde{Q}
\begin{bmatrix}
\xi \\
p_1(x,\xi) \\
\vdots \\
p_M(x,\xi) \\
f_1(W_1\xi) \\
\vdots \\
f_M(W_M\xi)
\end{bmatrix}
$$

$$
+\gamma \sum_{j=1}^{M} p_j(x,\xi)^\top p_j(x,\xi) + \theta \sum_{j=1}^{M} f_j^\top(W_j\xi) f_j(W_j\xi)
$$

$$
-\xi^\top \tilde{\Xi}^0 \xi - 2\sum_{j=1}^{M} \xi^\top W_j^\top \Gamma_j p_j(x,\xi) - 2\sum_{j=1}^{M} \xi^\top W_j^\top \Omega_j f_j(W_j\xi)
$$

$$
-2\sum_{j=1}^{M}\sum_{r=1}^{M} p_j(x,\xi)^\top W_j^\top W_j \tilde{\Upsilon}_{j,r} W_r^\top W_r f_r(W_r\xi).
$$

*Then, under Lemma 2, it can be deduced that*

$$
\dot{\tilde{V}} \leq 2\gamma \sum_{j=1}^{M} p_j(x,\xi)^\top p_j(x,\xi) + 2\theta \sum_{j=1}^{M} f_j^\top(W_j\xi) f_j(W_j\xi) - \xi^\top \tilde{\Xi}^0 \xi - 2\sum_{j=1}^{M} \xi^\top W_j^\top \Gamma_j p_j(x,\xi)
$$

$$
-2\sum_{j=1}^{M} \xi^\top W_j^\top \Omega_j f_j(W_j\xi) - 2\sum_{j=1}^{M}\sum_{r=1}^{M} p_j(x,\xi)^\top W_j^\top W_j \tilde{\Upsilon}_{j,r} W_r^\top W_r f_r(W_r\xi)
$$

$$
\leq -\xi^\top \tilde{\Xi}^0 \xi
$$

$$
-2\sum_{j=1}^{M} \xi^\top W_j^\top \left(\Gamma_j - \gamma L^j\right) p_j(x,\xi)
$$

$$
-2\sum_{j=1}^{M} \xi^\top W_j^\top \left(\Omega_j - \theta L^j\right) f_j(W_j\xi)
$$

$$
-2\sum_{j=1}^{M} p_j(x,\xi)^\top W_j^\top W_j \sum_{r=1}^{M} \left(\tilde{\Upsilon}_{j,r}\right) W_r^\top W_r f_r(W_r\xi).
$$

*Therefore, with the conditions equation 10, we can substantiate that the error dynamics of system equation 7 asymptotically approaches zero, meaning that the solution is contracting. Moreover, by Khalil & Grizzle (2002, Theorem 4.9). the stablility area of $\xi$ contains $\{\xi | V(\xi) \leq \max_{\zeta \in \mathbb{R}^d} \tilde{V}(\zeta)\}$. Since $y(t) = x(t) + \xi(t)$, we can determine that $\forall y_0 \in \{x_0 + \xi | V(\xi) \leq \max_{\zeta \in \mathbb{R}^d} \tilde{V}(\zeta)\}$, $y(t) \to x(t)$. This completes the proof.*

# C  MORE DISCUSSION ON RECONSTRUCTION ERROR OPTIMIZATION IN PRECISION-DEMANDING TASKS

Generative models utilize probabilistic modeling. The benefit of this approach is that once a distribution is learned, more new samples can be sampled, thus accomplishing the task of generation. However, in scenarios such as robot manipulation or temporal prediction, our primary objective is to model and understand the system, rather than approximating the underlying distribution and generating highly diverse and stylistically varied samples as is common in computer vision (CV). For instance, in imitation learning for robotic manipulation, expert human demonstration data often contains inherent jitter or other artifacts. Our objective is for the model to learn a smooth and robust policy, instead of modeling this underlying noise distribution. A mainstream framework in robotic

manipulation is to feed observed images as input and predict the robot's next action(s). This resembles a traditional supervised learning task (instead of probabilistic modeling), where we learn a mapping from inputs to outputs from the data. What differs is that the data distribution here often exhibits multi-modal properties. Taking the robot obstacle avoidance problem as an example: to avoid an obstacle ahead, the robot can detour by moving either left or right. These two options create a bimodal distribution in the data. If we simply use an Mean Squared Error (MSE) loss between the model's output and the collected sample data, the model may only learn an average behavior — ultimately causing it to move straight forward and collide with the obstacle. Therefore, Generative models such as diffusion (Chi et al., 2023) are used to capture this multi-modal nature in the data.

In summary, while traditional frequentist supervised learning enables us to discard the noise term and achieve high precision, probabilistic models allow us to capture more complex data distributions. But within each mode of the data distribution, we do not want to compromise on precision. In fact, the approach we adopt to conditional flow matching—via joint learning or optimal transport—achieves semantic segmentation of the noise space $x_0$ (Pooladian et al., 2023; Tong et al., 2024). Then our MLE fine-tuning based on the reconstruction results of each mode effectively compensates for the precision requirements in probabilistic models. The demand for precision necessitates the consideration of reconstruction error. This may also partially explain why VAE-based policy ACT (Zhao et al., 2023) has achieved significant success in robot fine manipulation tasks, despite being less capable than diffusion or flow-based models in capturing action diversity.

## D  EXPERIMENT DETAILS AND ADDITIONAL RESULTS

This section provides a detailed overview of the parameters used in our experiments to ensure reproducibility. All experiments were run on compute nodes, each node featuring eight A100 GPUs.

### D.1  CIFAR-10

We perform an experiment on unconditional CIFAR-10 generation from a Gaussian source to examine how OT-CFM performs in the high-dimensional image setting. We use a similar setup to that of Lipman et al. (2023) and Tong et al. (2024), including the time-dependent U-Net architecture from Nichol & Dhariwal (2021) that is commonly used in diffusion models. We choose channels $= 128$, depth $= 2$, channels multiple $= [1, 2, 2, 2]$, heads $= 4$, heads channels $= 64$, attention resolution $= 16$, dropout $= 0.1$, batch size per gpu $= 128$, gpus $= 4$, epochs $= 2000$. We use a constant learning rate, set to $2 \times 10^{-4}$ in the pre-training stage and $5 \times 10^{-6}$ in the fine-tuning stage. To prevent training instabilities and variance, we clip the gradient norm to 1 and rely on exponential moving average with a decay of 0.99.

For sampling, we use Euler integration using the torchdyn package and DOPRI5 from the torchdiffeq package. Since the DOPRI5 solver is an adaptive step size solver, it uses a different number of steps for each integration. We use a batch size of 1024 and average the number of function evaluations (NFE) over batches.

The visualization results of the generated images are presented in Fig. E.2. From Figs E.2b-E.2b, we can see that fine-tuning induces minimal alteration to the latent space semantics. Images generated from identical noise points retain consistent content, with a slight improvement in reconstruction fidelity (e.g., the 9th to 11th images from the end in the final row).

### D.2  ROBOTIC MANIPULATION

These experiments were conducted under a unified framework, systematically evaluated across multiple environments. The configuration encompasses four main aspects: model architectures, training hyperparameters, environment-specific parameters, and execution modes, ensuring reproducibility and consistency throughout the study.

We employ two types of network architectures to learn the vector field: a Conditional UNet-1D and a Transformer-based diffusion model. The UNet model was designed with an input dimension matching the action space of the environment and an output dimension corresponding to the action dimensions. It incorporated global conditioning based on visual feature encodings, a hidden size of 256, 4 UNet blocks with 2 layers each, and attention mechanisms to enhance representational capacity.

| H-Param | Ta | Tp | ObsRes | F-Net | F-Par | V-Enc | V-Par | P-Lr | F-Lr |
|---|---|---|---|---|---|---|---|---|---|
| Push-T | 8 | 16 | 1x96x96 | ConditionalUnet1D | 80 | ResNet-18 | 11 | 1e-4 | 5e-6 |
| Franka Kitchen | 8 | 16 | 1x60 | ConditionalUnet1D | 66 | N/A | N/A | 1e-4 | 5e-6 |
| Robomimic | 8 | 16 | 1x50 | ConditionalUnet1D | 66 | N/A | N/A | 1e-4 | 5e-6 |

Table D.1: Hyperparameters for FM and diffusion policy. Ta: action horizon. Tp: action prediction horizon. ObsRes: environment observation resolution. D-Net: diffusion/flow matching network. D-Par: diffusion/flow matching network number of parameters in millions. V-Enc: vision encoder. V-Par: vision encoder number of parameters in millions. P-Lr: learning rate in pretraining. F-Lr: learning rate in MLE fine-tuning.

| Tasks | Rob | Obj | ActD | PH | Steps | Img |
|---|---|---|---|---|---|---|
| Push-T | 1 | 1 | 2 | 200 | 300 | ✓ |
| Franka Kitchen | 1 | 7 | 9 | 566 | 280 | ✗ |
| Robomimic | 2 | 3 | 20 | 200 | 700 | ✗ |

Table D.2: Tasks Summary. Rob: number of robots. Obj: number of objects. ActD: action dimension. PH: proficient-human demonstration. Steps: max number of rollout steps. Franka Kitchen and Robomimic involve 6D robot and gripper actions in the joint space. Push-T focuses on robot end-effector trajectories.

The Transformer model used a hidden dimension of 512, 6 transformer layers, 8 attention heads, and a dropout rate of 0.1 to prevent overfitting. Both models utilized a ResNet-18 vision encoder for extracting image features, with GroupNorm substituted for BatchNorm to improve training stability.

We used a batch size of 64. In the pre-training stage the Adam optimizer was used with a learning rate of 1.0e-4 and weight decay of 1.0e-6, accompanied by a linear warm-up phase spanning 500 steps. While in MLE fine-tining stage we use a learning rate of 5.0e-6. In the design for inducing contraction properties we choose $\lambda_\omega = 0.05$ and $\epsilon_A = 10^{-6}$. All models were trained on 4 A100 GPUs using a Distributed Data Parallel strategy with 32-bit precision. An Exponential Moving Average was applied with a decay rate of 0.75, updated every 50 epochs.

Environment-specific parameters were carefully tailored to each experimental setting. For the kitchen environment, the observation horizon was set to 2, the prediction horizon to 24, and the action horizon to 8, with an action dimension of 9 and a visual feature dimension of 512. Each episode was allowed a maximum of 280 steps. In the Push-T environment, the observation horizon was 1, prediction horizon 16, and action horizon 8, with an action dimension of 2 and visual feature dimension of 514. The maximum steps per episode were 300. For the Mimic environment, the observation horizon was 1, prediction horizon 32, and action horizon 16, with action dimension 7 and visual feature dimension 512. Episodes were run for up to 400 steps.

The experiments supported several execution modes: FM pretraining mode; MLE fine-tuning mode or residual fine-tuning with a specialized network architecture; and testing mode with 10 test episodes, and 50 runs per episode by default. The maximum number of steps was automatically configured according to the environment settings. This modular configuration supports high customizability while ensuring the reproducibility of all experimental results.

Table D.1 shows the hyperparameters used in FM and diffusion policy. Table D.2 shows the task summary. Our evaluations were all conducted within the MuJoCo or Gym simulation environment. Specifically, the Push-T simulation environment is developed using Gym, whereas the Franka Kitchen and Robomimic environments utilize MuJoCo for simulation.

# E    THE USE OF LARGE LANGUAGE MODELS

We employed some Large Language Models (LLMs) to polish the writing and identify grammatical errors.

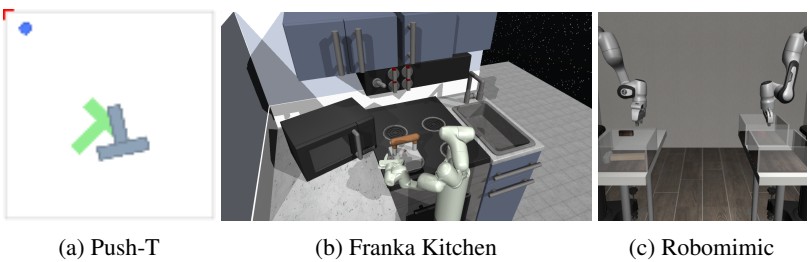

(a) Push-T          (b) Franka Kitchen          (c) Robomimic

Figure E.1: Different Simulation Environments in MuJoCo or Gym.

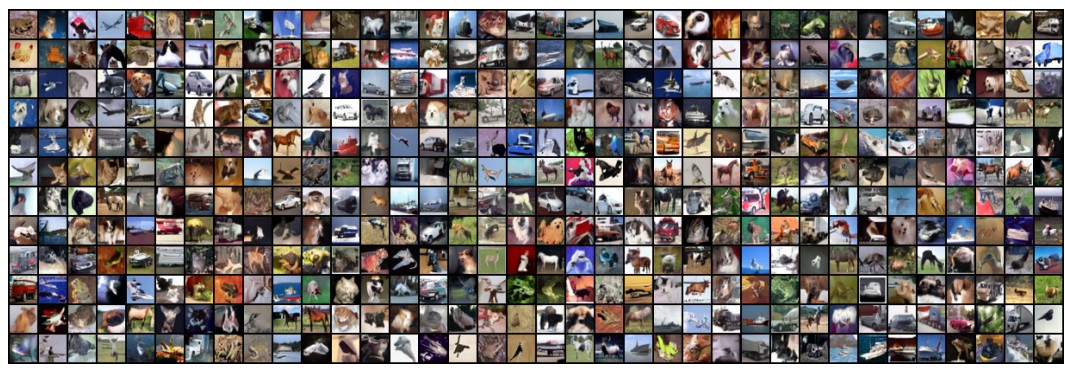

(a) Images generated by FM at $10^5$ step, FID: 6.17.

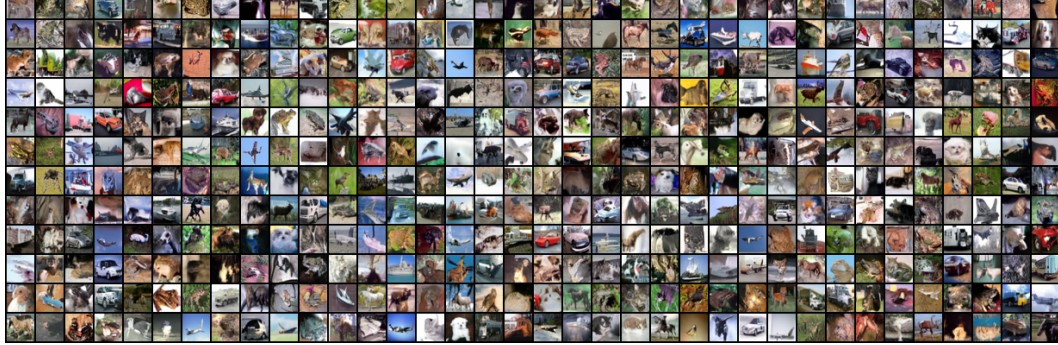

(b) Images generated by FM at $3.5 \times 10^5$ step, FID: 3.61.

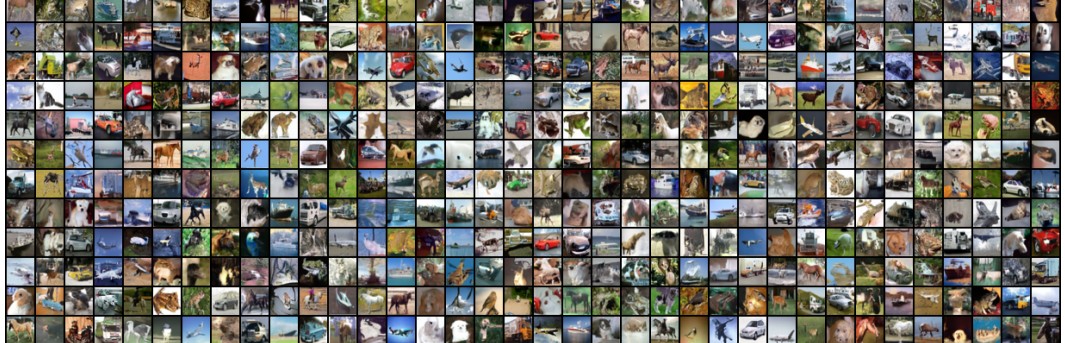

(c) Images generated by our fine-tuned FM model. Fine-tuning started from the $3.5 \times 10^5$ step and continued for $1,000$ steps, FID: 3.55.

Figure E.2: Visualization Results of generated images under the same initial noise points.

