# OpenReview forum: "Fine-Tuning Flow Matching via Maximum Likelihood Estimation of Reconstructions"
_ICLR.cc/2026/Conference — ICLR 2026 Conference Withdrawn Submission_

### Official Review · Reviewer_WNku · 2025-10-27

**Soundness:** 2
**Presentation:** 2
**Contribution:** 2
**Rating:** 4
**Confidence:** 3

**Summary:**

The authors target the train-inference gap in generative modeling via the flow-matching (FM) framework. They first offer a theoretical analysis connecting the training loss of FM (on the vector field) with actual inference error of the solved ODE. Building on this, they propose maximum-likelihood fine-tuning of the FM model by optimizing the likelihood of reconstructions rather than purely matching flow fields. Two fine-tuning paths are explored: (1) straightforward fine-tuning of the vector field, and (2) a residual-based fine-tuning architecture which incorporates a contraction property to improve robustness and interpretability. Experiments demonstrate that this fine-tuning improves alignment of generated samples with data reconstructions, closing the gap between training objectives and downstream performance. The work points toward more accurate and stable generative flows by combining classical maximum‐likelihood modeling with the continuous ODE structure of flow matching.

**Strengths:**

- The paper presents a novel perspective by linking flow matching to maximum likelihood estimation (MLE), offering a theoretically grounded fine-tuning objective that directly optimizes reconstruction likelihoods rather than the vector-field discrepancy. This connection between FM and MLE is original and provides new insight into improving generative model fidelity.

- The authors provide theoretical analysis clarifying the train–inference gap in flow-matching models and introduce two principled fine-tuning strategies, including a residual-based approach with contraction properties for better stability.

**Weaknesses:**

- The reported performance gains from MLE fine-tuning are relatively modest, especially compared to strong diffusion-based methods such as EDM. The paper would benefit from a deeper analysis explaining why the improvement remains limited, whether due to optimization dynamics, the expressiveness of the flow field, or dataset complexity.

 - The proposed approach has not been evaluated across different backbone architectures. Without such evidence, it is unclear whether the fine-tuning benefits generalize beyond the specific network used or if they depend heavily on architectural inductive biases (e.g., U-Net vs. Transformer). A brief analysis of architectural sensitivity would strengthen the claims of robustness.

- The choice to fine-tune an existing flow model rather than train from scratch using the MLE objective is not well justified. Exploring or at least discussing this design choice would clarify whether fine-tuning is essential for convergence stability, computational efficiency, or performance gains.

**Questions:**

- The reported improvement over baseline flow-matching models appears relatively minor, while existing diffusion-based methods such as EDM achieve stronger results. Could the authors discuss why the performance gain is limited?

- Is the proposed MLE fine-tuning method sensitive to the underlying network architecture (e.g., U-Net vs. Transformer backbones)? Have the authors tested whether architectural capacity or inductive bias affects the effectiveness of the fine-tuning procedure?

- Could the authors clarify why they chose to fine-tune a pre-trained flow-matching model instead of training one from scratch with the proposed MLE objective? Would training from scratch yield similar improvements, or is fine-tuning essential for stability or efficiency?

---

### Official Review · Reviewer_UjF6 · 2025-10-28

**Soundness:** 2
**Presentation:** 2
**Contribution:** 3
**Rating:** 4
**Confidence:** 3

**Summary:**

This paper proposes a Flow Matching (FM) fine-tuning method based on Maximum Likelihood Estimation (MLE) (FT-FM), aimed at improving the performance of generative models during the inference stage. The authors introduce a residual fine-tuning framework (ResFT-FM) to enhance model stability, and incorporate contraction theory and Linear Matrix Inequality (LMI) constraints to ensure stability. Experimental results validate the effectiveness of the FT-FM method in CIFAR-10 and robotic control tasks, improving generation quality and inference performance.

**Strengths:**

Originality:
  1. The paper exhibits clear originality in optimizing the Flow Matching (FM) model. By introducing Maximum Likelihood Estimation (MLE) to fine-tune the FM model, the authors provide a new approach to improve generative quality and stability. Compared to traditional generative model fine-tuning methods, the use of the residual fine-tuning framework (ResFT-FM) avoids overfitting and enhances model stability. These innovative approaches present a new direction in the flow matching and generative modeling fields.
 2. Additionally, the authors introduce contraction from control theory and Input-to-State Stability (ISS) analysis, providing strong theoretical support for the model's stability. This helps address the stability issues that generative models may encounter during the inference phase, significantly increasing the reliability of the model.

Quality:
  1. The experimental design is reasonable, and the theoretical analysis is thorough. However, there is room for improvement in the clarity of the derivations and the presentation of the figures.

Clarity:
  1. The writing is generally clear, but explanations of more complex sections could be improved, especially in terms of algorithmic steps and figure descriptions.
Significance:
The paper addresses the important problem of stability in generative models during the inference phase, and the methods proposed have significant implications for their application in robotic control tasks.

**Weaknesses:**

Lack of Sufficient Experimental Diversity:
 The experiments in the paper primarily focus on the CIFAR-10 image generation task and several robotic control tasks (e.g., Franka Kitchen and Push-T). While these tasks are somewhat representative, the experimental setup is relatively narrow and fails to demonstrate the method's applicability in other domains or broader tasks. There is a lack of validation on more diverse tasks and different datasets.

Insufficient Comparison with Other State-of-the-Art Models:
 Issue: Some of the experiments in the paper are compared with models from 2023 and earlier, with a lack of comparison to other generative models. This comparison might underestimate the limitations of the current method, especially as the field of generative models has seen significant advancements, and new methods might achieve notable improvements in performance.

Lack of Ablation Studies:
The paper does not include ablation studies, meaning there is no detailed validation of the impact of each component (e.g., residual fine-tuning, maximum likelihood estimation, control theory analysis) on the final performance. This makes it difficult for readers to understand the contribution of each individual part.

Insufficient Experimental Figures:
The paper has a limited number of figures, and the existing figures do not sufficiently compare different experimental conditions or models to showcase the method’s performance. Some of the figures lack adequate background explanations, and there is insufficient clarification of model origins and experimental settings.

Missing Algorithm Workflow:
The paper does not provide a detailed algorithm workflow or pseudocode, which could present challenges in understanding and reproducing the method. Providing clear algorithmic steps is particularly important when the method is complex.

**Questions:**

1. Issue: Training-Inference Gap Analysis
Question: Could you provide more details explaining how the training-inference gap analysis in the paper differs from or relates to the previous work in Error Bounds for Flow Matching Methods (Benton et al., 2023)? Is there a new contribution, particularly in theoretical derivation or experimental validation?
Suggestions:
Clarify Innovation: If Benton et al. (2023) have already made a similar analysis, the authors should explicitly point out the differences between their work and prior research, especially in how the training-inference gap is addressed from a new perspective or method.
Cite Relevant Literature: To avoid potential redundancy, it would be helpful for the authors to provide more detailed citations of existing literature in the background section and explain how their work differs from or improves upon previous studies. This will help to clarify the paper's originality and contribution.

2. Experimental Setup and Task Diversity
Question:
The current experiments focus on CIFAR-10 and several robotic tasks (e.g., Franka Kitchen and Push-T). The experimental setup is relatively narrow and does not showcase the method's applicability in other types of tasks. Specifically, in generative modeling and robotic control, there is a lack of comparison with the latest generative models.
Suggestions:
Could the authors provide experiments across a broader range of tasks (e.g., text generation, image super-resolution) to validate the method’s applicability? Furthermore, could comparisons with more cutting-edge or advanced generative models be added in future versions?

3. Ablation Study
Question:
The paper lacks detailed ablation studies, and does not demonstrate the independent impact of different components (e.g., residual fine-tuning, maximum likelihood estimation, and control theory analysis) on model performance.
Suggestions:
We encourage the authors to include more ablation studies, particularly comparing FT-FM and ResFT-FM by removing components like residual fine-tuning or control theory analysis. This would help clarify the contribution of each component to the final performance and further validate the method’s effectiveness.

4. Clarity of Algorithm Workflow
Question:
The paper does not provide a detailed algorithm workflow or pseudocode, and lacks a clear explanation of each step, which may affect reproducibility and understanding, particularly for complex methods and fine-tuning frameworks.
Suggestions:
We suggest that the authors provide a detailed workflow of the algorithm, potentially using pseudocode or flowcharts to illustrate each key step. This would improve the reproducibility of the paper and help readers better understand the implementation details.

5. Performance on Larger-Scale Tasks
Question:
The paper validates the method's effectiveness on small-scale experiments, but there is a lack of testing in larger-scale or real-world applications, especially on large datasets or complex robotic tasks.
Suggestions:
Could the authors further demonstrate the method’s performance on large-scale datasets (e.g., ImageNet) or more complex robotic control tasks (e.g., long-duration, high-precision tasks)? This would provide additional validation of the method's feasibility and superiority in real-world applications.

---

### Official Review · Reviewer_VZgV · 2025-11-01

**Soundness:** 2
**Presentation:** 3
**Contribution:** 2
**Rating:** 4
**Confidence:** 4

**Summary:**

This paper proposes to finetune flow matching with MLE of reconstruction. The paper argues that the flow matching lacks supervision from reconstruction like VAE, NF and GAN. Therefore, for high precision tasks like robotic control, the error gap between training and sampling is vulnerable. Furthermore, the flow matching target the straightness which lead to the stiff of system and discontinuous vector field.

**Strengths:**

1. This paper is well-organized and easy to follow.
2. The technique boosts the performance of robot tasks in table 2

**Weaknesses:**

1. The paper argues that flow matching targets at the straightness making the learnable vector field discontinuous, and it would rather be like SDE. However, given flow matching, we can still apply the corresponding SDE formula [1] to do sampling. Have the author tested on this SDE sampling ?

[1]: SiT: Exploring Flow and Diffusion-based Generative Models with Scalable Interpolant Transformers

2. If the flow matching is not suitable because of following the straightness, then one possible solution to using stochastic diffusion model to model robotic task. There is a paper training diffusion with MLE objective [2]. How [2] performs compared to proposed technique.

[2]: Maximum Likelihood Training of Score-Based Diffusion Models

3. Another way to increase the precision of sampling is to use higher order sampling like DPM [3], this could boost the precision of sampling. However, the paper lacks comparison with the advance sampling algorithm.

[3]: DPM-Solver: A Fast ODE Solver for Diffusion Probabilistic Model Sampling in Around 10 Steps

4. The proposed technique does not boost performance well on image generation tasks like Cifar10. Table 1 does not show significant support to the paper proposed technique.

**Questions:**

See the weaknesses above

---

### Note · Authors · 2025-12-07

I have read and agree with the venue's withdrawal policy on behalf of myself and my co-authors.